# Thymosin α1 protects from CTLA-4 intestinal immunopathology

Giorgia Renga[1],*, Marina M Bellet[1],*, Marilena Pariano[1], Marco Gargaro[1], Claudia Stincardini[1], Fiorella D'Onofrio[1], Paolo Mosci[1], Stefano Brancorsini[1], Andrea Bartoli[1], Allan L Goldstein[2] (ORCID), Enrico Garaci[3], Luigina Romani[1] (ORCID), Claudio Costantini[1] (ORCID)

The advent of immune checkpoint inhibitors has represented a major boost in cancer therapy, but safety concerns are increasingly being recognized. Indeed, although beneficial at the tumor site, unlocking a safeguard mechanism of the immune response may trigger autoimmune-like effects at the periphery, thus making the safety of immune checkpoint inhibitors a research priority. Herein, we demonstrate that thymosin α1 (Tα1), an endogenous peptide with immunomodulatory activities, can protect mice from intestinal toxicity in a murine model of immune checkpoint inhibitor–induced colitis. Specifically, Tα1 efficiently prevented immune adverse pathology in the gut by promoting the indoleamine 2,3-dioxygenase (IDO) 1–dependent tolerogenic immune pathway. Notably, Tα1 did not induce IDO1 in the tumor microenvironment, but rather modulated the infiltration of T-cell subsets by inverting the ratio between CD8$^+$ and Treg cells, an effect that may depend on Tα1 ability to regulate the differentiation and chemokine expression profile of DCs. Thus, through distinct mechanisms that are contingent upon the context, Tα1 represents a plausible candidate to improve the safety/efficacy profile of immune checkpoint inhibitors.

## Introduction

Cancer immunotherapy, defined as the fourth pillar of human cancer therapy, next to surgery, chemotherapy, and radiotherapy, aims to coopt the immune system to combat cancer and puts down its roots back in 1890 when Coley proved that injection of bacteria or bacterial lysates promoted remission in patients with malignant sarcomas (Szeto & Finley, 2019). Despite these early results, the development of cancer immunotherapy was delayed by the prevailing use of strategies aimed at combating the tumor cells directly

rather than promoting an inflammatory response (Ritter & Greten, 2019). However, it soon became clear that the tumor microenvironment played a fundamental role in tumor development, reinvigorating the attention on the immune component, but it was only recently that immunotherapy gained a central place in cancer therapy with the development of immune checkpoint inhibitors (ICIs) (Kelly, 2018; Ritter & Greten, 2019). Since an excessive activation of the immune system might be detrimental, immune checkpoints work as brakes to keep the immune response under control, thus playing a fundamental role for the proper functioning of the immune system. This physiological mechanism, however, may be coopted by tumors as a strategy to elude the surveillance of the immune system and prevent the onset of an antitumor immune response. The use of ICI serves this purpose. Indeed, by removing the brakes imposed by the tumor on immune cells, it is expected that a vigorous antitumor immune response will be established with potential curative effects. Clinical application of ICI encountered an immediate success; however, it soon became clear that they are often not curative, and there are several types of cancer that are resistant to the therapy (National Academies of Sciences, Engineering, and Medicine, 2019). This led to the idea that the use of immune checkpoint blockade should be combined to other cancer therapies to increase the effectiveness against cancer (National Academies of Sciences, Engineering, and Medicine, 2019). In addition, the clinical application of ICI is not only hampered by a limited efficacy but also by the occurrence of immune-related side effects that could undermine its safety (Martins et al, 2019), among which diarrhea and/or colitis are common adverse events (Samaan et al, 2018). Therefore, combination with other molecules/treatments preserving the mucosal barrier integrity should also be actively pursued to improve the safety of ICI.

ICI-induced colitis shares endoscopic and histological features with inflammatory bowel disease and the treatment of choice is represented by corticosteroids, followed by infliximab (anti-TNFα antibody) and vedolizumab (anti α4β7 integrin antibody) in steroid-

---

[1]Department of Experimental Medicine, University of Perugia, Perugia, Italy    [2]Department of Biochemistry and Molecular Medicine, School of Medicine and Health Sciences, The George Washington University, Washington, DC, USA    [3]University San Raffaele and Istituto di Ricovero e Cura a Carattere Scientifico (IRCCS) San Raffaele, Rome, Italy

Correspondence: luigina.romani@unipg.it
*Giorgia Renga and Marina M Bellet contributed equally to this work

and steroid/infliximab–refractory patients, respectively (Som et al, 2019). However, it remains unexplored whether any measure could be taken to prevent, rather than cure, ICI-induced colitis (Som et al, 2019). One possibility relies on coopting endogenous mechanisms of tolerance and protection as a strategy to restore or maintain mucosal homeostasis (Porter et al, 2018), which may include immune cell types, such as regulatory T cells or tolerogenic DCs, and cytokines (Porter et al, 2018).

Thymosin α1 (Tα1), an N-terminal acetylated acidic peptide of 28 amino acids first isolated from the thymic tissue (Goldstein et al, 1977), has been long characterized for its immunomodulatory activities (King & Tuthill, 2016) linked to the promotion of tolerogenic indoleamine 2,3-dioxygenase 1 (IDO1) activity at mucosal surfaces (Romani et al, 2006, 2017; Montagnoli et al, 2008). Based on these premises, in the present study, we have assessed whether Tα1 could prevent or ameliorate gastrointestinal toxicity in a mouse model of

ICI-induced colitis. We found that Tα1 protects mice from anti–CTLA-4–induced colitis by engaging the IDO1 tolerogenic pathway in the gut, while sustaining the antitumor activity via DCs and lymphocyte infiltration at the tumor site. These results suggest the potential use of Tα1 in mitigating immune side effects associated with checkpoint inhibitors blockade.

# Results

## Tα1 protects against ICI-induced gastrointestinal toxicity

We first asked whether Tα1 could protect against gastrointestinal toxicity induced by dextran sulfate sodium (DSS). We found that Tα1 partially prevented weight loss (Fig 1A) and increased survival of DSS-treated mice (Fig 1B). This was associated with an increased

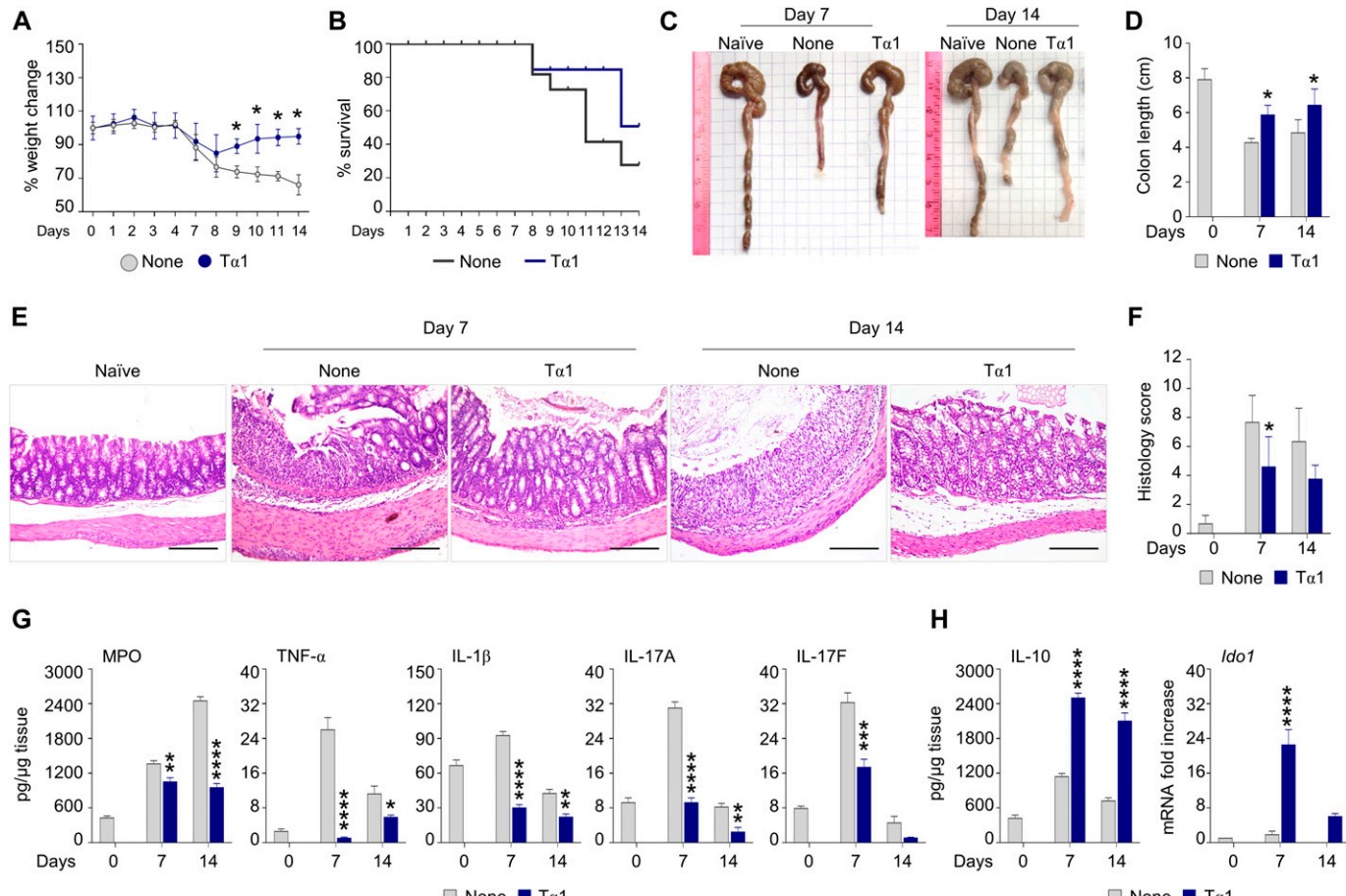

**Figure 1. Thymosin α1 (Tα1) protects mice from DSS-induced colitis.**
C57BL/6 mice were subjected to DSS-induced colitis for 1 wk followed by a recovery period of another week. Fresh DSS solution was added at day +3. Tα1 was administered every other day at the dose of 200 μg/kg, intraperitoneally. **(A, B, C, D, E, F, G, H)** Mice were evaluated for (A) % weight change, (B) % survival, (C) gross pathology, (D) colon length (cm), (E) colon histology (hematoxylin and eosin staining), (F) histology score, (G) levels of inflammatory cytokines, and (H) IL-10 in colon homogenates and *Ido1* expression. Cytokines were determined by ELISA and gene expression was performed by RT-PCR (data are presented as mean ± SD of three independent experiments). Images were taken with a high-resolution microscope (Olympus BX51), 20× magnification (scale bars, 200 μm). For histology, data are representative of three independent experiments. Each in vivo experiment includes four mice per group. Weight change and survival are calculated on a total of 12 mice per group. *P < 0.05, **P < 0.01, ***P < 0.001, ****P < 0.0001, Tα1-treated versus untreated (DSS) mice. Two-way ANOVA, Bonferroni or Tukey's post hoc test. None, mice with DSS colitis only. Naïve, untreated mice.

colon length (Fig 1C and D), reduced inflammatory pathology (Fig 1E and F), decreased production of myeloperoxidase (MPO), TNF-α, IL-1β, IL-17A, and IL-17F (Fig 1G) and increased levels of IL-10 (Fig 1H). Of interest, Tα1 also promoted IDO1 mRNA expression in the colon (Fig 1H).

Because DSS-induced colitis is primarily mediated by innate immune mechanisms with a limited contribution of adaptive immunity (Kiesler et al, 2015) and the anti–CTLA-4 treatment is expected to induce a vigorous T cell response, we asked whether the protective effect of Tα1 could be still present in the DSS plus anti–CTLA-4 model, in which a concerted action of innate and adaptive immunity contributes to gastrointestinal toxicity. For this purpose, we resorted to the recently described murine model of ICI-induced colitis in which a combination of DSS and anti–CTLA-4 antibody has been used (Wang et al, 2018, 2019; Perez-Ruiz et al, 2019). As shown in Fig 2, Tα1 administration prevented weight loss (Fig 2A), increased survival (Fig 2B), and improved disease activity score (Fig 2C) of DSS plus anti–CTLA-4–treated mice. This was associated with an improved gross pathology (Fig 2D). Although colon length was not increased (Fig 2E and F), Tα1 ameliorated colon histopathology (Fig 3A and B) and restored epithelial barrier integrity by inducing the proliferation of intestinal stem cells and preventing the loss of epithelial cells, as shown by Ki-67 and TUNEL staining, respectively (Fig 3C). Consistent with the improved barrier integrity, Tα1 prevented the passage of dextran-FITC (Fig 3D) and of *Candida albicans*, a gut commensal that disseminates upon disruption of the barrier integrity

(Fig 3E). These results indicate that Tα1 protects from gastrointestinal toxicity in a mouse model of ICI-induced colitis by counteracting the inflammatory pathology and providing mucosal homeostasis.

## IDO1 is instrumental for the protective activity of Tα1

To get insights into the mechanism by which Tα1 protects against gastrointestinal toxicity, we evaluated the local expression and activity of IDO1, known to be induced by Tα1 (Romani et al, 2006, 2017; Montagnoli et al, 2008) and to promote mucosal homeostasis in the gut (Romani et al, 2017). As shown in Fig 4, the amounts of IDO1 mRNA (Fig 4A) and protein (Fig 4B) as well as the levels of kynurenine/tryptophan ratio (Fig 4C) were increased by Tα1, indicating the engagement of tryptophan catabolism along the kynurenine pathway. Accordingly, the tryptophan levels decreased (Fig 4C). Consistent with the high levels of IL-10 at the effector sites (Fig 4D), the expression of regulatory T cells (Treg) markers *Foxp3* and *Ikaros* (Thomas et al, 2019) in the mesenteric lymph nodes was also increased by Tα1 (Fig 4E), whereas the levels of proinflammatory IL-1β and IL-17A were decreased (Fig 4D), a finding indicating the promotion of a tolerogenic immunoprotective pathway.

Subsequent studies in IDO1-deficient mice proved that IDO1 function is causally linked to the protective activity of Tα1. In agreement with previous studies (Shon et al, 2015), IDO1-deficient mice were not more susceptible to DSS-induced colitis (Fig 4F and

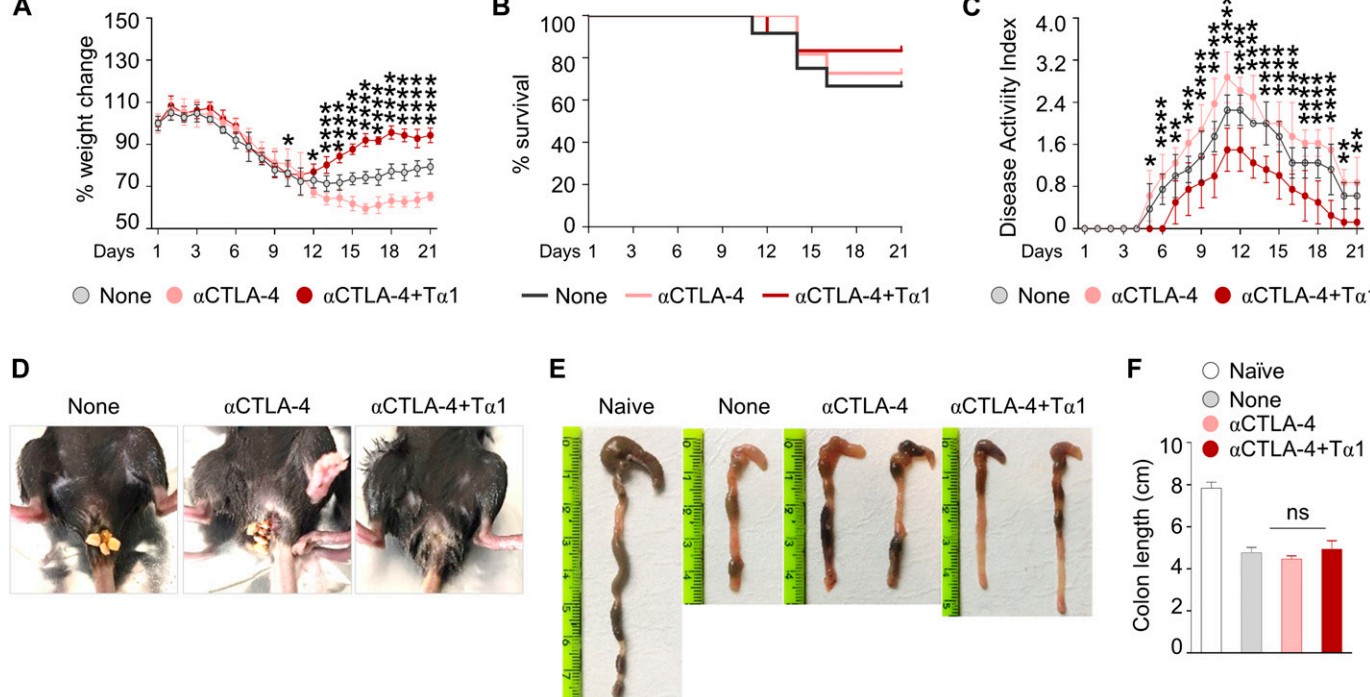

**Figure 2. Thymosin α1 (Tα1) protects mice from DSS plus anti–CTLA-4–induced colitis.**
C57BL/6 mice were subjected to DSS-induced colitis for 1 wk followed by a recovery period of another week and administered 100 μg of anti–CTLA-4 mAb or isotype control twice (at days 0, 4, and 8 after DSS administration). Tα1 was administered every other day at a dose of 200 μg/kg, intraperitoneally. **(A, B, C, D, E, F)** Mice were evaluated for (A) % weight change, (B) % survival, (C) disease activity index, (D) rectal bleeding, (E) gross pathology, and (F) colon length (cm). Each in vivo experiment includes four mice per group. Weight change and survival are calculated on a total of 12 mice per group. *P < 0.05, **P < 0.01, ***P < 0.001, ****P < 0.0001, Tα1-treated versus untreated (DSS plus anti–CTLA-4 only) mice. Two-way ANOVA, Bonferroni or Tukey's post hoc test. None, mice with DSS colitis only. Naïve, untreated mice.

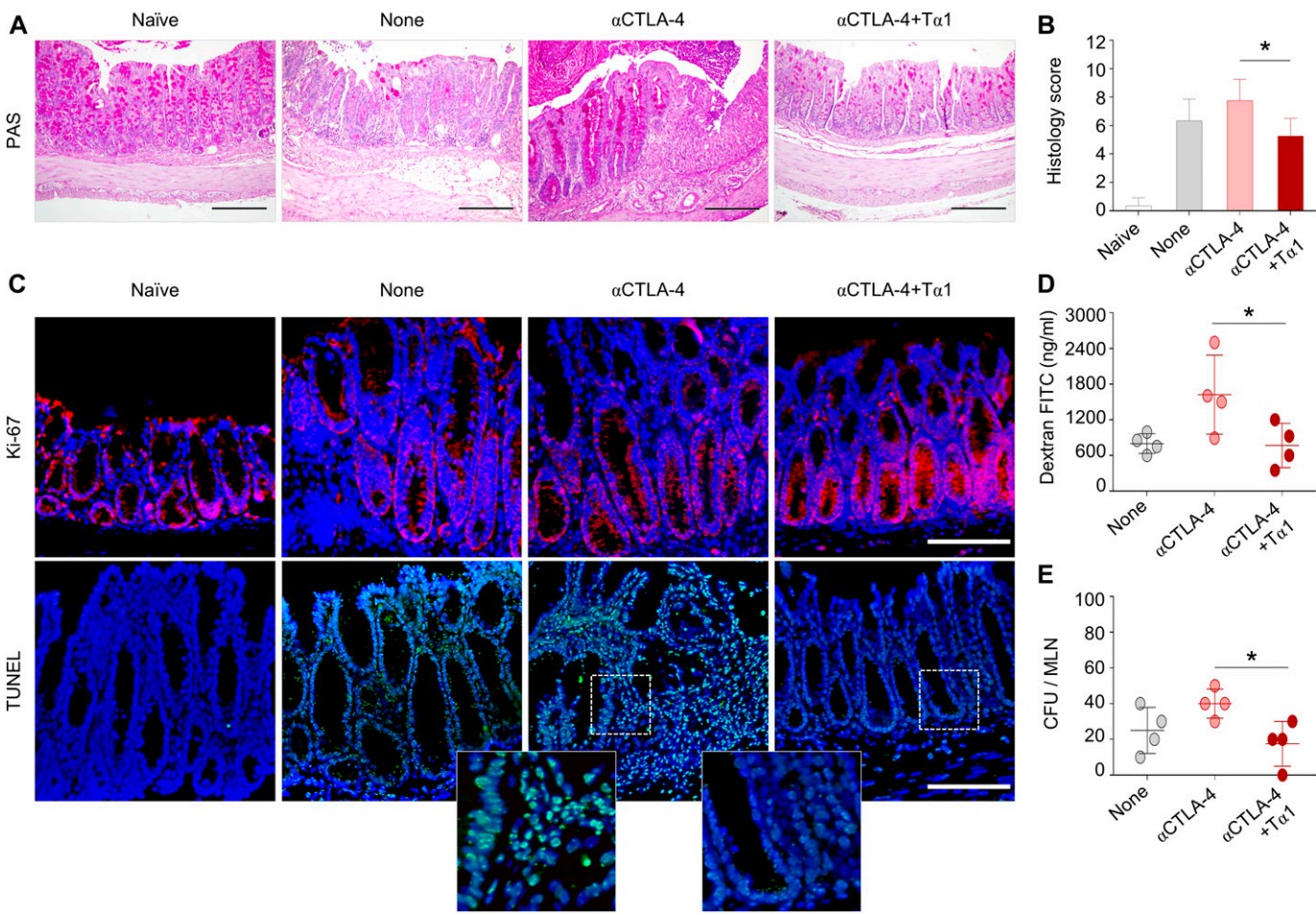

**Figure 3. Thymosin α1 (Tα1) prevents epithelial damage in DSS plus anti–CTLA-4–induced colitis.**

C57BL/6 mice were subjected to DSS plus anti–CTLA-4–induced colitis and administered Tα1 as described in the legend of Fig 2. **(A, B, C, D, E)** Mice were evaluated for (A) colon histology (periodic acid-Schiff staining), (B) histology score, (C) Ki-67 expression and TUNEL staining, (D) dextran-FITC levels in the serum, and (E) fungal growth ($\log_{10}$ CFUs) in the mesenteric lymph nodes. For immunofluorescence, nuclei were counterstained with DAPI. Photographs were taken with a high-resolution microscope (Olympus BX51), 20× magnification (scale bars, 200 $\mu m$). For histology and immunofluorescence, data are representative of two independent experiments. In vivo experiment includes four mice per group. *$P < 0.05$, Tα1-treated versus untreated (DSS plus anti–CTLA-4 only) mice. One-way ANOVA, Bonferroni or Tukey's post hoc test. None, mice with DSS colitis only. Naïve, untreated mice.

G). However, their susceptibility increased upon concomitant administration of anti–CTLA-4 (Fig 4F and G). Of note, the protective effects of Tα1 were abolished in the absence of IDO1, as indicated by unaltered colon pathology (Fig 4F and G) and unaltered expression of *Il10* and *Foxp3* (Fig 4H). Confirming these results, neither Foxp3+ cells nor Rorγt+ cells were affected in the colon upon Tα1 treatment (Fig 4F). These results indicate that Tα1 engages IDO1 in the gastrointestinal tract to protect against ICI-induced colitis by inducing an anti-inflammatory and tolerogenic pathway.

### Tα1 may potentiate the antitumor activity of anti–CTLA-4

A potential application of Tα1 in combination therapy with ICI requires that the protective activity in the gastrointestinal tract occur without interference with the antitumor effects. To prove this, we treated mice with B16 melanoma with anti–CTLA-4 antibody, with and without Tα1. As expected, anti–CTLA-4 significantly reduced the tumor growth (Fig 5A), while promoting tumor necrosis as revealed

by dark pigmented cells (dead tumor cells had increased expression of melanin and appeared as darkly pigmented cells) and TUNEL assay (Fig 5B). Neither activity was antagonized by the concomitant treatment with Tα1. Actually, Tα1 promoted the infiltration of CD8+ T cells penetrating the viable and nonviable tumor tissue (Fig 5B), a positive prognostic factor for the efficacy of ICI, and increased the expression of CD8 associated markers (*GzmB* and *Perforin*) (Fig 5C). Flow cytometry revealed that the combination of the anti–CTLA-4 antibody and Tα1 increased the frequency of CD8+ and CD4+ T cells at the tumor site (Fig 5D and E). Among CD4+ T cells, Tα1, alone or in combination with anti–CTLA-4 antibody, significantly reduced the number of Foxp3+CD25+ Treg cells (Fig 5F and G). These data suggest that Tα1 may shape the tumor environment by selectively influencing T-cell infiltration.

As opposed to the gut, Tα1 did not induce IDO1 mRNA (Fig 5H) and, actually, decreased kynurenine production in the tumor mass while the tryptophan levels were slightly increased (Fig 5I). Accordingly, the Kyn/Trp ratio was also significantly reduced (Fig 5I).

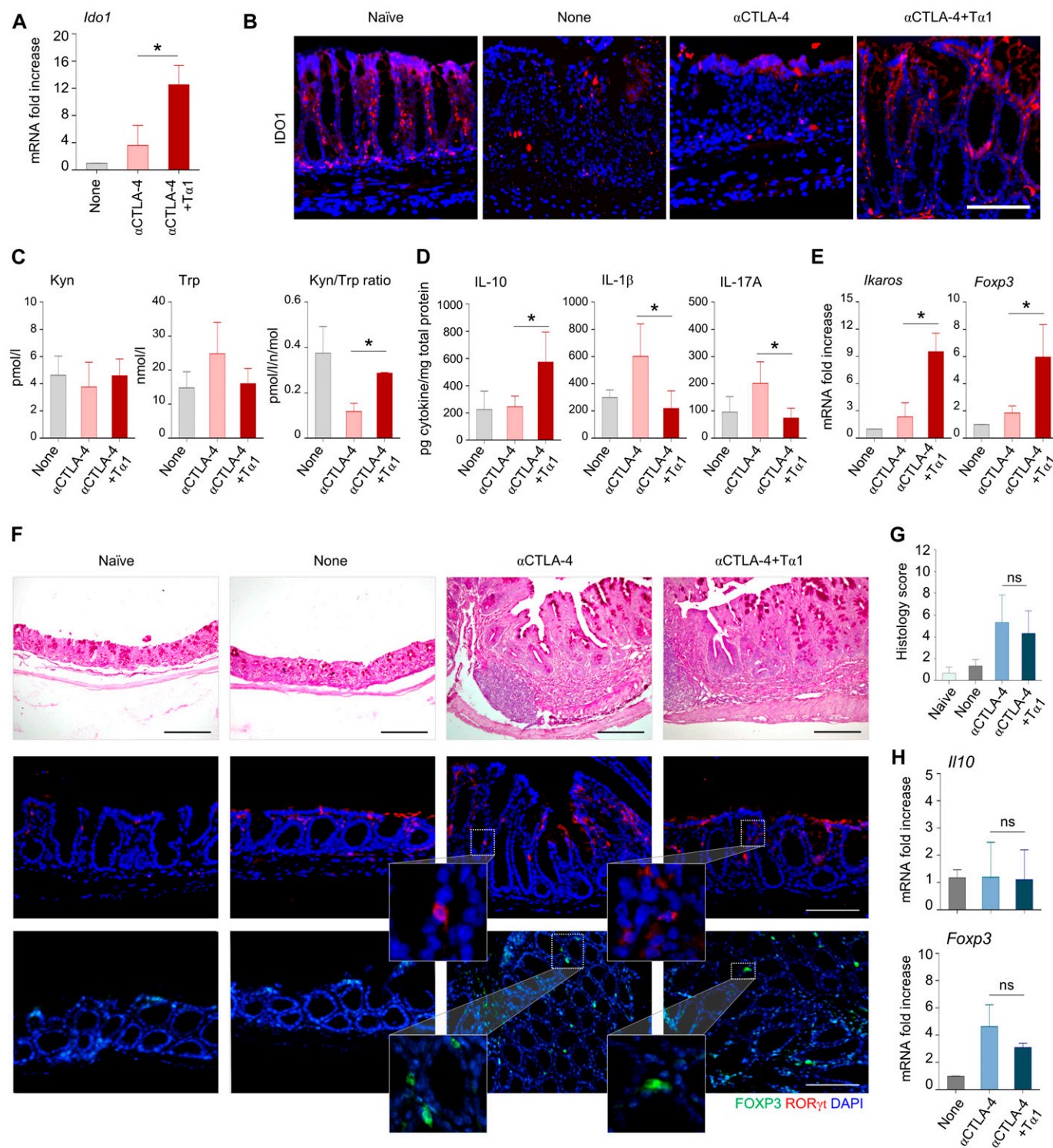

**Figure 4. Thymosin α1 (Tα1) prevents epithelial damage in DSS plus anti–CTLA-4–induced colitis via IDO1.**
C57BL/6 or *Indo*$^{-/-}$ mice were subjected to DSS plus anti–CTLA-4–induced colitis and administered Tα1 as described in the legend of Fig 2. **(A, B, C, D, E)** C57BL/6 mice were evaluated for (A) *Ido1* gene and (B) IDO1 protein expression, (C) kynurenine (Kyn), tryptophan (Trp) levels, and Kyn/Trp ratio, (D) IL-10, IL-1β and IL-17A levels in colon homogenates, and (E) *Ikaros* and *Foxp3* expression in mesenteric lymph node. **(F, G, H)** *Indo*$^{-/-}$ mice were evaluated for (F) colon histology (periodic acid-Schiff staining) and expression of Foxp3- and Roryt-positive cells, (G) histology score and (H) *Il10* and *Foxp3* expression in colon. Cytokines were determined by ELISA and gene expression was performed by RT-PCR (data are presented as mean ± SD or ± SEM of three independent experiments). For immunofluorescence, nuclei were counterstained with DAPI. Photographs were taken with a high-resolution microscope (Olympus BX51), 20× (scale bars, 200 μm) and 40× magnification (scale bars, 100 μm) for histology and immunofluorescence, respectively. For histology and immunofluorescence, data are representative of two independent experiments. In vivo experiment includes four mice per group. *P < 0.05, Tα1-treated versus untreated (DSS plus anti–CTLA-4 only) mice. *t* test or one-way ANOVA, Bonferroni or Tukey's post hoc test. None, mice with DSS colitis only. Naïve, untreated mice.

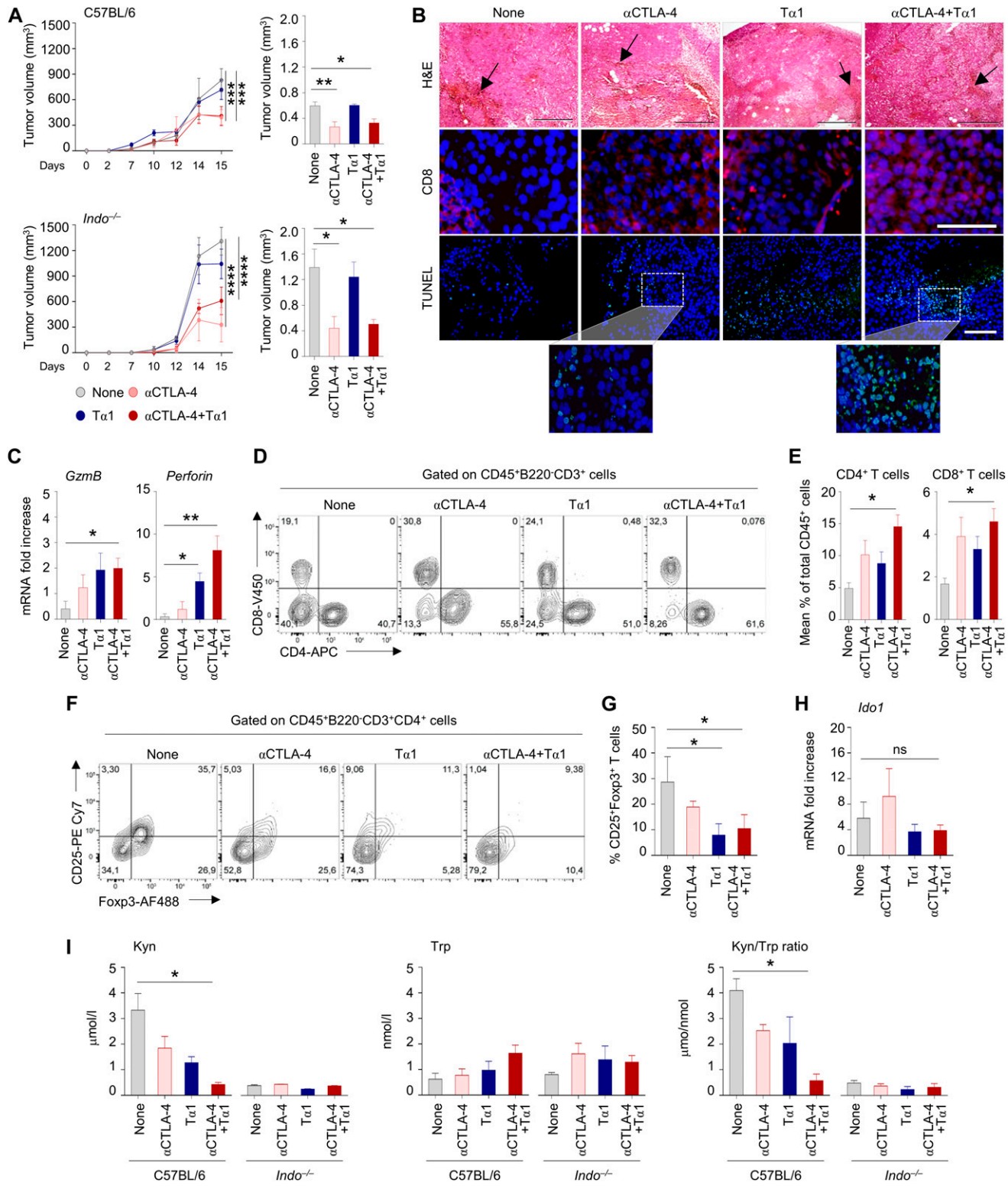

**Figure 5. Thymosin α1 (Tα1) preserves anti–CTLA-4 antitumor activity in melanoma.**

C57BL/6 or *Indo*[−/−] mice were subcutaneously injected with B16 tumor cells and administered 100 μg anti–CTLA-4 mAb or isotype control intraperitoneally four times at 3-d intervals up to 15 d. Tα1 was administered every other day at a dose of 200 μg/kg, intraperitoneally. **(A, B, C, D, E, F, G, H, I)** Mice were evaluated for (A) tumor growth, (B) histology (H&E staining), CD8[+] cells infiltration and tumor cell death (TUNEL), (C) local expression of *GzmB* and *Perforin*, (D, E) frequency of CD8[+]CD4[+] T cells and (F, G) CD25[+]Foxp3[+] T cells in tumor homogenates, quantified by flow cytometry (H) *Ido1* gene and (I) kynurenine (Kyn), tryptophan (Trp) levels and Kyn/Trp ratio. Gene expression was performed by RT-PCR (data are presented as mean ± SD of two independent experiments). Tumor growth data are presented as mean ± SEM of two

Experiments in IDO1-deficient mice validated these findings. In agreement with previous studies (Holmgaard et al, 2013), the absence of IDO1 did not prevent tumor growth, but increased the efficacy of anti–CTLA-4 antibody (Fig 5A) that was maintained in the presence of Tα1 (Fig 5A). These data indicate that Tα1 may be used in combination therapy with anti–CTLA-4 antibody because of its ability to uncouple the antitumor activity of anti–CTLA-4 from gut immunotoxicity.

These findings were confirmed in a different tumor setting. Specifically, we intravenously injected C57BL/6 mice with Lewis lung carcinoma cells as a model of orthotopic lung adenocarcinoma (Janker et al, 2018) and treated mice with anti–PD1 antibody (Li et al, 2017), in the presence or absence of Tα1. As expected, anti-PD1 antibody reduced tumor growth as indicated by the decreased lung weight (Fig S1) and reduced gross pathology (Fig S1). Tα1 neither impairs the antitumor activity (Fig S1) nor the gross pathology improvement of anti-PD1 (Fig S1). As observed in the melanoma tumor model, Tα1 increased the expression of CD8-associated markers (*GzmB* and *Perforin*) (Fig S1) while reducing the recruitment of Foxp3⁺CD25⁺ Treg cells in the lung in combination with anti-PD1 antibody (Fig S1). These results suggest that Tα1 not only does not interfere with the antitumor activity of checkpoint inhibitors, but actually modulates the tumor microenvironment to favor their efficacy.

### Tα1 modulates the chemokine profile to promote tumor lymphocyte infiltration

To investigate the mechanism by which Tα1 promotes the differential recruitment of T-cell subsets, we measured the levels of chemokines in the melanoma tumor microenvironment and found increased levels of *Cxcl9* and *Cxcl10*, known to promote tumor lymphocyte infiltration, and decreased levels of *Ccl22*, implicated in intratumoral recruitment of Treg, upon treatment with anti–CTLA-4 antibody and Tα1 (Fig 6A). It has been recently shown that innate immune sensing of tumors occurs through DC activation and regulation of tumor lymphocyte infiltration via production of CXCL9 and CXCL10 (Spranger et al, 2017). Interestingly, we found an increased percentage of MHCII⁺CD11c⁺ DC expressing CXCL9 in the tumor mass upon administration of anti–CTLA-4 antibody and Tα1 (Fig 6B and C), suggesting that Tα1 can indeed modify the chemokine profile at the tumor site likely by regulating the expression program of DCs.

As a matter of fact, macrophages were not modified by the treatment with Tα1 (Fig S2).

Our group had already shown that Tα1 is able to promote DC subsets generation, activation, and cytokine production by signaling through Toll-like receptors (Romani et al, 2004, 2006; Bozza et al, 2007; Yao et al, 2007; Perruccio et al, 2010). Here, we further characterized primary cultures of bone marrow cells exposed to Tα1

(Tα1-DC) in comparison with bone marrow cells differentiated with GM-CSF/IL-4 (GM-DC) or FLT3 ligand (FLT3L; FL-DC). Analyzed by light and electron microscopy for morphological appearance, GM-DC and FL-DC displayed a different morphology, with GM-DC being larger and with more abundant cytoplasm (Fig 6D), in agreement with previous findings (Xu et al, 2007). Tα1-DC, in contrast, showed intermediate size compared with GM-DC and FL-DC, with the presence of numerous short pseudopods, as revealed by electron microscopy (Fig 6D). The cells were then characterized for surface phenotype by assessing CD11c, B220, and CD11b expression (Fig 6E). As expected (Naik et al, 2005; Xu et al, 2007; Helft et al, 2015), most GM-DCs were negative for B220, so did not contain any plasmacytoid dendritic cell (pDC) and expressed high levels of CD11b, whereas FL-DCs expanded both B220⁺ pDCs and conventional dendritic cells (Fig 6E). Surface phenotype of Tα1-DCs revealed the presence of both B220⁺ pDCs and CD11bʰⁱᵍʰ myeloid cells, similar to FL-DCs (Fig 6E). Consistent with the ability of Tα1 to signal through TLR (23), known to induce differentiation of myeloid progenitors (Nagai et al, 2006; McGettrick & O'Neill, 2007; Downes & Marshall-Clarke, 2010), the promotion of DC by Tα1 was mostly TLR9/TRIF-dependent (Fig 6F). On performing a microarray analysis of cytokines, chemokines, and receptors (Fig S3), we found that, among others, CXCL10 was up-regulated in Tα1-DC as compared with GM-DC and FL-DC, whereas CCL22 was apparently down-regulated (Fig 6G), consistent with the results obtained at the tumor site. Thus, Tα1 seems to promote the differentiation of DC with a chemokine profile that may be relevant within the tumor microenvironment.

## Discussion

The results presented in this study reveal an intriguing, yet crucial role of Tα1 in antitumor immunotherapy that may bring the power of ICI to more oncology patients. Tα1 promoted the CD8⁺ cell infiltration at the tumor site, a prerequisite for ICI efficacy, while adversing abscopal immunotoxicity at distal sites. Since its purification from thymic extracts more than 40 yr ago and the proven ability to activate T-cell differentiation and function (Goldstein et al, 1977), there has always been an interest for the potential application of Tα1 in cancer therapy (Costantini et al, 2019). The general consensus emerging from the literature is that Tα1 may be efficacious against a variety of tumors, especially when used in combination with other immune- or chemotherapies, with an excellent safety profile (Costantini et al, 2019). However, by coupling with the distinct mechanisms through which ICI provide antitumor activity as well as immune adverse events, Tα1 may represent a possible ICI partner candidate. Indeed, the use of ICI poses an additional level of complexity related to the unrestrained activation of T cells endowed with autoimmune-like systemic effects. Efforts have been recently published in order to increase the efficacy of ICI

independent experiments. Black arrows in histology sections indicate the presence of dark pigmented cells. For immunofluorescence, nuclei were counterstained with DAPI. Photographs were taken with a high-resolution microscope (Olympus BX51), 10× magnification (scale bars, 500 μm) for histology, and TUNEL assay, 40× magnification (scale bars, 100 μm) for immunofluorescence. For histology and immunofluorescence, data are representative of two independent experiments. In vivo experiment includes 3–6 mice per group. *P < 0.05, **P < 0.01, ***P < 0.001, ****P < 0.0001, Tα1-treated versus untreated (anti–CTLA-4) mice. t test or one-way ANOVA, Bonferroni or Tukey's post hoc test. None, untreated mice.

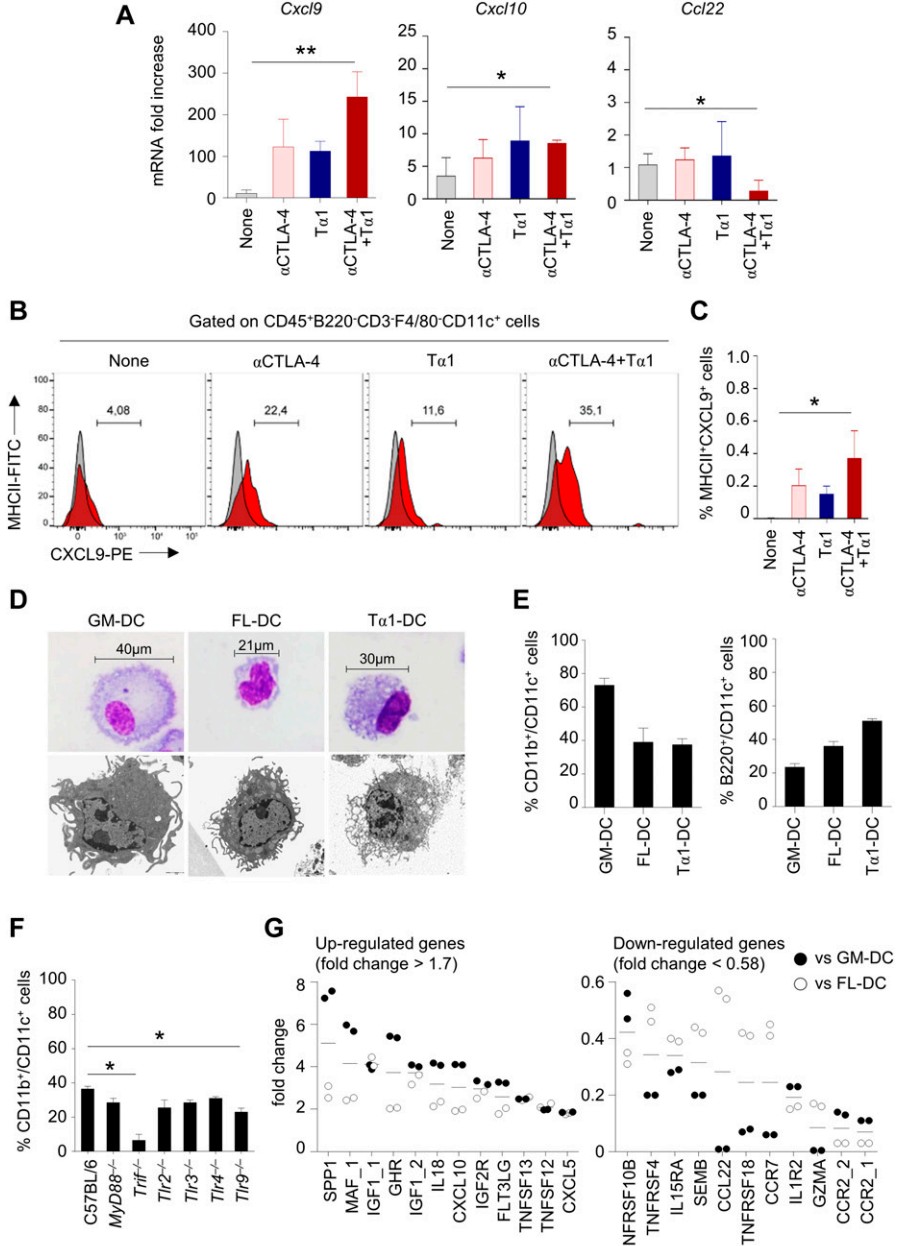

**Figure 6. Thymosin α1 (Tα1) induces the differentiation of DC from bone marrow precursors with a specific gene expression profile.**
**(A)** Chemokine gene expression in C57BL/6 mice with B16 melanoma, treated with 100 μg anti–CTLA-4 mAb with and without Tα1 as in legend to Fig 5. Gene expression was performed by RT-PCR. Data are presented as mean ± SD of two independent experiments. **(B, C)** Frequency of MHCII⁺-CXCL9⁺ cells in tumor homogenates, quantified by flow cytometry. **(D, E)** Bone marrow precursor cells were treated with Tα1 alone (Tα1-DC), GM-CSF/IL-4 (GM-DC), or FLT3 ligand (FL-DC) and evaluated for (D) morphology by light and electron microscopy (representative images of two independent experiments) and (E) expression of CD11c, B220, and CD11b by flow cytometry. **(F)** Percentage of CD11b⁺/CD11c⁺ cells obtained from primary cultures of bone marrow cells prepared from mice deficient of selected TLRs as well as associated adapters and exposed to Tα1 evaluated by flow cytometry. **(G)** Gene expression in DC by microarray. The genes up-regulated, or down-regulated, in Tα1-DC compared to both GM-DC (black dots) and FL-DC (white dots) are indicated. Each dot represents a biological replicate.

at the tumor site while minimizing immunotoxicity. For instance, engineering anti–CTLA-4 to selectively localize its activity at the tumor site prevented peripheral toxicity (Pai et al, 2019) and treatment with TNF inhibitors concomitantly with CTLA-4 and PD1 antibodies ameliorated colitis while improving antitumor efficacy (Perez-Ruiz et al, 2019). Tα1 seems to work along the same line. Tα1 resorted to IDO1 to limit the ICI-mediated immunotoxicity at the mucosal site. The ability of Tα1 to engage the IDO1 pathway is not surprising. Indeed, we have previously shown that Tα1 promoted the activation of tolerogenic DC capable to mediate antimicrobial immunity and alloantigen tolerization in hematopoietic transplanted mice via IDO1 (Romani et al, 2006). Similarly, Tα1 improved the inflammatory phenotype and promoted immune tolerance via

IDO1 in a murine model of cystic fibrosis (Romani et al, 2017). Collectively, these results indicate that Tα1 engages the IDO1 pathway in experimental conditions characterized by excessive inflammation to mitigate the immune response and promote immune homeostasis. Based on this assumption, it is not surprising that IDO1 is not induced by Tα1 at the tumor site, considering that B16 cells, although not expressing IDO1 (Holmgaard et al, 2015), are potent inducers of IDO1 in other cell types (Sharma et al, 2007; Holmgaard et al, 2013). Indeed, we did not observe increased levels of kynurenines or IDO1 expression at the tumor site upon Tα1 treatment. IDO1 is considered a negative prognostic factor in tumors because of its ability to induce an immune-suppressed environment despite the failure of a recent clinical trial using an

IDO1-selective enzyme inhibitor in combination with an anti-PD1 antibody in advanced melanoma (Muller et al, 2019). In line with a previous study (Holmgaard et al, 2013), we did not observe any change in B16 tumor growth in wild-type and IDO1-deficient mice, suggesting that IDO1 is not a prerequisite for tumor growth. Irrespective of the specific functions of IDO1 in the tumor microenvironment, we did not observe any modulation of IDO1 activity by Tα1, suggesting that Tα1 may activate IDO1 in conditions of over-activated immune response. The property to react in a context-dependent manner may not come as a surprise, given the moonlighting activity of Tα1, including its ability to activate different pattern recognition receptors and down-stream signaling pathways (Romani et al, 2012). This demands for other mechanisms mediating Tα1 activity at the tumor site. We found that Tα1 promoted the tumor infiltration of CD8[+] cells, thus increasing the tumor immunoscore, now considered a marker of improved overall survival in response to ICI (Pages et al, 2018; Kumpers et al, 2019; Angell et al, 2020). The mechanisms at the basis of these effects may include the production of CXCL10, a chemokine recognized for its role in the recruitment of tumor-infiltrating lymphocytes being required for antitumor immune responses following immune checkpoint blockade (House et al, 2020) and acting as a biomarker for long-term survival of melanoma patients (Kaesler et al, 2019). Indeed, not only DC differentiated from bone marrow precursors in the presence of Tα1 expressed high levels of CXCL10 but CXCL10 was also expressed in the tumor microenvironment in response to Tα1. In contrast, CCL22, abundantly expressed in many types of cancer and instrumental for intratumoral recruitment of Treg (Anz et al, 2015), was down-regulated by Tα1 in DC and at the tumor site. Thus, Tα1 seems to promote the differentiation of DC with a chemokine profile that may be relevant within the tumor microenvironment. Moreover, given the ability of Tα1 to regulate MHC class I expression (Giuliani et al, 2000), it is likely that an increased antigenicity of tumors could facilitate recognition by cytotoxic CD8[+] T cells.

In conclusion, our results indicate that Tα1 may be used in combination therapy with ICI to improve their safety profile and likely promoting their antitumor efficacy via distinct pathways that work to normalize the immune response at the tumor and peripheral sites.

# Materials and Methods

### Mice, models and treatments

C57BL/6 mice were purchased from Charles River Laboratories. B6129indo (Indo1[−/−]), MyD88[−/−], Trif[−/−], Tlr2[−/−], Tlr3[−/−], Tlr4[−/−], and Tlr9[−/−] mice were bred under specific pathogen-free conditions in the Animal Facility of Perugia. 5- to 8-wk-old male and female mice were used in all experiments. Murine experiments were performed according to Italian Approved Animal Welfare Authorization 360/2015-PR and Legislative Decree 26/2014 regarding the animal license obtained by the Italian Ministry of Health lasting for 5 yr (2015–2020). In the DSS colitis model, mice received 3% DSS (MP Biomedicals) in their drinking water for 7 d with or without DSS refilling at day +3. Weight was recorded daily. Mice were injected intraperitoneally with 100 μg of anti–CTLA-4 mAb (clone 9D9-BioXCell) or isotype control (clone MPC-11-BioXCell) twice (at days 0, 4, and 8 after the DSS administration). In the melanoma model, mice were subcutaneously injected into the right flank with 2 × 10[5] B16 tumor cells. Mice were injected four times at 3-d intervals with 100 μg of isotype control or anti–CTLA-4 mAb. Tumor size, expressed in mm[3], was measured by a caliper for 15 d after tumor inoculation. Tumor volume was determined every 2–3 d after inoculation (width^2 × length/2)*1,000 and at sacrifice (length × width × height). In the Lewis lung carcinoma model, mice were intravenously injected with 2 × 10[5] LLC1 tumor cells. Mice were injected five times at 3-d intervals with 200 μg of isotype control or anti-PD1 mAb (clone RMP1-14-BioXCell) and euthanized 18 d after tumor cell inoculation. Tα1 was supplied as purified (the endotoxin levels were <0.03 pg/ml, by a standard limulus lysate assay), sterile, lyophilized, acetylated polypeptide. The sequence was as follows: Ac-Ser-Asp-Ala-Ala-Val-Asp-Thr-Ser-Ser-Glu-Ile-Thr-Thr-Lys-Asp-Leu-Lys-Glu-Lys-Lys-Glu-Val-Val-Glu-Glu-Ala-Glu-Asn-O. In both the DSS colitis and tumor models, Tα1 was administered every other day at the dose of 200 μg/kg intraperitoneally.

### Clinical signs and histopathology scores

The severity of colitis was assessed by calculating disease activity index. All mice were monitored for stool consistency and rectal bleeding daily as previously described (Wirtz et al, 2017). Briefly, stool scores were determined as follows: 0 = well-formed pellets, 1 = semi-formed stools that did not adhere to the anus, 2 = semi-formed stools that adhered to the anus, and 3 = liquid stools that adhered to the anus. Bleeding scores were determined as follows: 0 = no blood, 1 = positive hemoccult, 2 = blood traces in stool visible, and 3 = gross rectal bleeding. For histological evaluations, colonic sections were examined and scored in a blinded fashion to avoid any bias. Based on the existing literature (Engel et al, 2011), four histological components were assessed: "inflammation extent," "damage in crypt architecture," "hyperemia/edema," and "grade of accumulation with inflammatory cells." The colonic sections were scored from 0 to 3 points for each parameter. The total histological score, ranging from 0 to 12, was obtained by summing the four histological components' scores.

### Immunofluorescence

The tissues were removed and fixed in 10% phosphate-buffered formalin (Bio Optica), embedded in paraffin and sectioned at 3 μm. For histological analysis, sections were stained with periodic acid-Schiff or hematoxylin and eosin reagents. For immunofluorescence, the sections were rehydrated and, after antigen retrieval in citrate buffer (10 mM, pH 6), fixed in 2% formaldehyde for 40 min at room temperature and permeabilized in a blocking buffer containing 5% FBS, 3% BSA, and 0.5% Triton X-100 in PBS. The slides were then incubated at 4°C with primary antibodies anti-IDO1 (clone 10.1; Millipore), anti–Ki-67 (Abcam), anti-FOXP3 (clone 150D; BioLegend), anti-RORγt (clone REA278; Miltenyi), and anti-CD8 (clone 5H10-1; Cell Signaling Technology). After extensive washing with PBS, the slides were then incubated at room temperature for 60 min with secondary antibodies, anti-mouse Alexa Fluor 555 (Thermo Fisher

Scientific), and anti-Rabbit TRITC (BETHYL). Nuclei were counterstained with DAPI. Images were acquired using a microscope BX51 and analySIS image processing software (Olympus).

## TUNEL staining

Sections were deparaffinized, rehydrated, and treated with 0.1 M citrate buffer, pH 6.0, for 20 min in a water bath, washed, and fixed in 4% buffered paraformaldehyde, pH 7.3, for 36 h. The sections were then washed and blocked in 0.1 M Tris/HCl buffer, pH 7.5, and supplemented with 3% bovine serum albumin and 20% FCS. The slides were then incubated with fluorescein-coupled dUTP and TUNEL enzyme (Roche Diagnostics) in the presence of terminal deoxynucleotidyltransferase. The samples were then washed with PBS, incubated for 10 min at 70°C to remove unspecific binding. The sections were mounted and analyzed by fluorescent microscopy using a 20× objective.

## Intestinal permeability

Intestinal permeability was measured in fasted C57BL/6 mice for 4 h before the administration of 40 mg/100 g mouse weight of FITC-dextran (4 kD; Sigma-Aldrich) as described (Chen et al, 2008). Serum was collected retro-orbitally 4 h later and diluted 1:3 in PBS. The amount of fluorescence at 488 nm for emission and absorption at 525 nm, was read on the Infinite 200 plate reader (Tecan) using the manufacturer's I-control version 1.3 software. *Candida* growth in mesenteric lymph nodes was expressed as $\log_{10}$ CFU, obtained by serially diluting homogenates on Sabouraud agar plates incubated at 37°C for 24 h.

## Kynurenine and tryptophan assay

IDO1 functional activity was measured in vitro in terms of the ability to metabolize tryptophan to kynurenine whose concentrations were measured by using competitive ELISA kits according to the manufacturer's instructions (Labor Diagnostika Nord).

## Flow cytometry analysis

For the melanoma model, tumors were isolated from mice and digested with Collagenase IV and DNase (Sigma-Aldrich) in HBSS for 30 min at 37°C with agitation and filtered through a 70-$\mu$m cell strainer to make a single cell suspension. For the Lewis lung carcinoma model, lungs containing orthotopic tumors were harvested, minced with scissors, and digested with Collagenase P (Sigma-Aldrich) and DNase in HBSS for 30 min at 37°C. The total cell suspension was resuspended in FACS analysis buffer and then stained with the following antibodies for 30 min at 4°C in the dark: V450-conjugated anti-CD8 (clone 53-6.7; BD Horizon), APC-conjugated anti-CD4 (Miltenyi), PE Cy7–conjugated anti-CD25 (clone PC61; BD Pharmingen), AF488-conjugated anti-Foxp3 (clone 150D; BioLegend), FITC-conjugated anti-MHCII (clone M5/114.15.2; BioLegend), PE-conjugated anti-CXCL9 (clone MIG-2F55; BioLegend), AF700-conjugated anti-CD11b (clone M1/70; BioLegend), BV711-conjugated anti-F4/80 (clone T45-2342; BD Horizon), APC Cy7-conjugated anti-CD45 (clone 104; BD Pharmingen), PE Cy7-conjugated anti-B220 (clone RA-6B2;

BioLegend), SB600-conjugated anti-CD3 (clone 145-2C11; Thermo Fisher Scientific), and BV650-conjugated anti-CD11c (clone HL3; BD Horizon). Intracellular staining was conducted using the Cytofix/Cytoperm plus kit (BD PharMinigen). After staining, the cells were washed with FACS PBS and quantified using the BD LSRFortessa cell analyzer (Becton Dickinson). Gating strategy has been shown in Fig S4.

## DC subset generation and cultures

GM-DC or FL-DC were obtained from bone marrow cells cultured for 7–9 d in the presence of recombinant GM-CSF (rGM-CSF; Schering-Plough) and rIL-4 (Peprotech, Inalco) or FLT3L (R&D Systems), as described (Romani et al, 2006). T$\alpha$1-DC were obtained by addition of T$\alpha$1 (20 $\mu$g/ml) for 5 d. After differentiation, $10^6$ differentiated cells were resuspended in FACS analysis buffer and stained as previously described (Romani et al, 2006).

## Morphological analysis

GM-DC, FL-DC, and T$\alpha$1-DC were centrifuged at room temperature onto slides at 100,000 per slide. Slides were air-dried and stained with May-Grunwald-Giemsa for morphological analysis. Observations were made by means on an inverted microscope at 400× magnification.

## Electron microscopy

Collected cells were fixed in cacodylate fixative buffer (0.1 M sodium cacodylate, 2% paraformaldehyde, and 3% glutaraldehyde) overnight at 4°C. The cells were then washed with 0.2 M sodium cacodylate buffer and dehydrated on an alcohol series (30%, 50%, 70%, 80%, 90%, and 100%) for 15 min each. Specimens were then embedded into acrylic resin. Ultrafine sections were obtained by cutting into the resin specimens with a glass blade on an ultra-microtome and mounted on nickel grids. Grids were stained with 2% uranyl acetate. Micrographs were taken with an EM 208 transmission electron microscope (Phillips).

## Microarray

Gene expression analysis was performed using topic-defined PIQOR Cytokine & Receptors Microarrays by MACSmolecular Genomics Services (Miltenyi Biotec GmbH). Briefly, RNA was isolated using standard RNA extraction protocols (NucleoSpin RNA II; Macherey-Nagel). Integrity of total RNA was evaluated using the Agilent Bioanalyzer 2100 system (Agilent Technologies). RNA integrity number was between 6.7 and 7.8, and thus, samples were considered suitable for further processing. 1 $\mu$g of each total RNA sample was used for the linear T7-based amplification step and amplified RNA checked with Agilent Bioanalyzer 2100 system. Samples were then labeled according to the PIQOR User Manual, and fluorescently labeled samples were hybridized overnight to topic-defined PIQOR Cytokine & Receptors Microarrays Mouse Antisense using the a-Hyb Hybridization Station. Fluorescent signals of the hybridized PIQOR Microarrays were detected using Agilent's DNA microarray scanner (Agilent Technologies). Mean

signals and mean local background intensities were obtained for each spot of the microarray images using the ImaGene software (Biodiscovery). Low-quality spots were flagged and excluded from data analysis. Unflagged spots were analyzed with the PIQOR Analyzer software.

## ELISA

Murine MPO, IL-1$\beta$, IL-10, IL-17A, IL-17F, and TNF-$\alpha$ cytokine concentration was determined in organ homogenates by using specific ELISA kits according to the manufacturers' instructions (eBioscience Inc., R&D System and BioLegend).

## Real-time PCR

Real-time PCR was performed using the CFX96 Touch Real-Time PCR detection system and iTaq Universal SYBR Green Supermix (Bio-Rad). Organs were lysed and total RNA was extracted using TRIzol Reagent (Thermo Fisher Scientific) and reverse transcribed with PrimeScript RT Reagent Kit with gDNA Eraser (Takara), according to the manufacturer's directions. Amplification efficiencies were validated and normalized against $\beta$-actin. The thermal profile for SYBR Green real-time PCR was at 95°C for 3 min, followed by 40 cycles of denaturation for 30 s at 95°C and an annealing/extension step of 30 s at 60°C. Each data point was examined for integrity by analysis of the amplification plot. The mRNA-normalized data were expressed as relative mRNA levels with respect to control. The following murine primers were used: *β-actin*: forward AGCCATG-TACGTAGCCATCC, reverse CTCTCAGCTGTGGTGGTGAA; *Ccl22*: forward CTGATGCAGGTCCCTATGGT, reverse GCAGGATTTTGAGGTCCAGA; *Cxcl9*: forward ACGGAGATCAAACCTGCCT, reverse TTCCCCCTCTTTTGCTTTTT; *Cxcl10*: forward AAGTGCTGCCGTCATTTTCT, reverse CCTATGGCCCT-CATTCTCAC; *Foxp3*: forward CCCAGGAAAGACAGCAACCTTTT; reverse TTCTCACAACCAGGCCACTTG; *GzmB*: forward CTCTGCCTTCTTCCTCTCCT, reverse CCAGAGACAAGGTCAGCAGT; *Ido1*: forward CCCACACTGAGCACGGACGG, reverse GCCCTTGTCGCAGTCCCCAC; *Ikaros*: forward AGCGGGGAG-CAGATGAAGGTGTA, reverse CGTACCGGTCCTGGCTGTGG; *Il10*: forward GAGAAGCATGGCCCAGAAATCAAG, reverse ATCACTCTTCACCTGCTC-CACTGC; *Perforin*: forward GGTGGACTGACAAGATGGAC, reverse CTCACATGTCACCTCATGGA.

## Statistical analysis

*t* test, one-way, and two-way ANOVA with Bonferroni post hoc test were used to determine the statistical significance. Significance was defined as $P < 0.05$. Data are pooled results (mean ± SD, mean ± SEM) or representative images from three (for the DSS colitis model) or two experiments (for the melanoma model). The in vivo groups consisted of 3–6 mice/group. GraphPad Prism software 6.01 (GraphPad Software) was used for analysis.

# Supplementary Information

# Acknowledgements

We thank Cristina Massi Benedetti for digital art and editing. The light and electron microscopy images were realized by Bonifazi Pierluigi during his PhD thesis.

## Author Contributions

G Renga: data curation, formal analysis, validation, investigation, and methodology.
MM Bellet: data curation, formal analysis, validation, investigation, and methodology.
M Pariano: data curation, formal analysis, investigation, and methodology.
M Gargaro: investigation and methodology.
C Stincardini: formal analysis, investigation, and methodology.
F D'Onofrio: investigation and methodology.
P Mosci: investigation and methodology.
S Brancorsini: formal analysis.
A Bartoli: formal analysis.
AL Goldstein: formal analysis.
E Garaci: formal analysis.
L Romani: conceptualization and formal analysis.
C Costantini: conceptualization and formal analysis.

## Conflict of Interest Statement

A patent application on "Thymosin $\alpha$1 for use in treatment of checkpoint inhibitor immune adverse events" by G Renga, MM Bellet, M Pariano, C Costantini, E Garaci, and L Romani is pending (IT201900016310).

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
