## [Reviewer comments · Life Science Alliance]

Thymosin alpha 1 protects from CTLA-4 intestinal immunopathology

Giorgia Renga, Marina Bellet, Marilena Pariano, Marco Gargaro, Claudia Stincardini, Fiorella D'Onofrio, Paolo Mosci, Stefano Brancorsini, Andrea Bartoli, Allan Goldstein, Enrico Garaci, Luigina Romani and Claudio Costantini

DOI: 10.26508/lsa.202000662

Corresponding author(s): Prof. Luigina Romani (University of Perugia)

Review timeline:

Submission Date:	2020-01-28
Editorial Decision:	2020-03-10
Revision Received:	2020-07-17
Editorial Decision:	2020-08-05
Revision Received:	2020-08-07
Accepted:	2020-08-07

Transaction Report:

No Peer Review Process File is available with this article, as the authors have chosen not to make the review process public in this case.

1st Editorial Decision

10 March 2020

Re: Life Science Alliance manuscript #LSA-2020-00662

Prof. Luigina Romani
University of Perugia
P.le Gambuli, 06132 Perugia, Italy
Perugia 06132
Italy

Dear Dr. Romani,

Thank you for submitting your manuscript entitled "Thymosin alpha 1 protects from CTLA-4 intestinal immunopathology" to Life Science Alliance. The manuscript was assessed by expert reviewers, whose comments are appended to this letter.

As you will see, the reviewers appreciate your analyses. However, they also think that more support for your conclusions is needed. They provide constructive input on how to provide such support and we would like to invite you to submit a revised version of your manuscript to us, addressing the individual reviewer concerns. Importantly, additional support for the efficacy of tumor inhibition in presence of thymosin alpha1 is needed as well as a better analysis of DC and T cells. Furthermore, quantifications and statistical tests need to get performed.

Thank you for this interesting contribution to Life Science Alliance. We are looking forward to receiving your revised manuscript.

Sincerely,

Andrea Leibfried, PhD
Executive Editor
Life Science Alliance

Meyerhofstr. 1
69117 Heidelberg, Germany
t +49 6221 8891 502
e a.leibfried@life-science-alliance.org
www.life-science-alliance.org

B. MANUSCRIPT ORGANIZATION AND FORMATTING:

2nd Editorial Decision

5 August 2020

August 5, 2020

RE: Life Science Alliance Manuscript #LSA-2020-00662R

Prof. Luigina Romani
University of Perugia
P.le Gambuli, 06132 Perugia, Italy
Perugia 06132
Italy

Dear Dr. Romani,

Thank you for submitting your revised manuscript entitled "Thymosin alpha 1 protects from CTLA-4 intestinal immunopathology". We would be happy to publish your paper in Life Science Alliance pending final revisions necessary to meet our formatting guidelines.

- please take a look at our manuscript preparation guidelines and order your manuscript sections accordingly
- please add a conflict of interest statement to your main manuscript text
- please list 10 authors et al. in your reference list
- please add scale bars to Figure 2B
- please check your figures S3 & S4 and make sure that the panels in the figure are in the figure legends and the main manuscript text

To upload the final version of your manuscript, please log in to your account: <https://lsa.msubmit.net/cgi-bin/main.plex>

A. FINAL FILES:

-- Summary blurb (enter in submission system): A short text summarizing in a single sentence the study (max. 200 characters including spaces). This text is used in conjunction with the titles of papers, hence should be informative and complementary to the title. It should describe the context and significance of the findings for a general

readership; it should be written in the present tense and refer to the work in the third person. Author names should not be mentioned.

B. MANUSCRIPT ORGANIZATION AND FORMATTING:

We encourage our authors to provide original source data, particularly uncropped/-processed electrophoretic blots and spreadsheets for the main figures of the manuscript. If you would like to add source data, we would welcome one PDF/Excel file per figure for this information. These files will be linked online as supplementary "Source Data" files.

Sincerely,

Reilly Lorenz
Editorial Office Life Science Alliance
Meyerhofstr. 1
69117 Heidelberg, Germany
t +49 6221 8891 414
e contact@life-science-alliance.org
www.life-science-alliance.org

3rd Editorial Decision

07 August 2020

RE: Life Science Alliance Manuscript #LSA-2020-00662RR

Prof. Luigina Romani
University of Perugia
P.le Gambuli, 06132 Perugia, Italy
Perugia 06132
Italy

Dear Dr. Romani,

Thank you for submitting your Research Article entitled "Thymosin alpha 1 protects from CTLA-4 intestinal immunopathology". It is a pleasure to let you know that your manuscript is now accepted for publication in Life Science Alliance. Congratulations on this interesting work.

*****IMPORTANT: If you will be unreachable at any time, please provide us with the email address of an alternate author. Failure to respond to routine queries may lead to unavoidable delays in publication.*****

DISTRIBUTION OF MATERIALS:

Again, congratulations on a very nice paper. I hope you found the review process to be constructive and are pleased with how the manuscript was handled editorially. We look forward to future exciting submissions from your lab.

Sincerely,

Reilly Lorenz
Editorial Office Life Science Alliance
Meyerhofstr. 1

Life Science Alliance

Life Science Alliance - Peer Review Process File

69117 Heidelberg, Germany
t +49 6221 8891 414
e contact@life-science-alliance.org
www.life-science-alliance.org